# Seismicity catalogue of the entire Chilean margin (18° to 56°S) from an automated approach

Martin Riedel-Hornig[1], Christian Sippl[2], Andrés Tassara[1], Sergio Ruiz[3], Bertrand Potin[3], Jorge Puente[2]

[1]Departamento de Ciencias de la Tierra, Universidad de Concepción, Concepción, Chile

[2]Institute of Geophysics, Czech Academy of Sciences, Prague, Czech Republic

[3]Departamento de Geofísica, Universidad de Chile, Santiago, Chile

*Correspondence to*: Martin Riedel-Hornig (mriedel@udec.cl)

**Abstract.** In this study we process 5 years of continuous seismic data, from 2017 to 2021, recorded by networks deployed
along the Chilean convergent margin of western South America. We select a set of stations so that coverage is as continuous and consistent in time as possible. Thus, changes in the catalogue are mainly due to the processes responsible for the seismicity itself and not network artifacts. For phase picking, we use the deep-learning algorithm Earthquake Transformer, PyOcto for association, NonLinLoc for location and HypoDD for relocation. To handle the large spatial extent of the study area, we create a partitioning workflow that mitigates problems that arise from associator and location codes working in cartesian coordinates.
This allows us to obtain a main seismicity catalogue with over 600.000 double-difference relocated events and a completeness magnitude of ~2.5, with events located using an existing 3D velocity model. While this provides better location accuracy than a 1D or 2D velocity model, it limits the final catalogue to the extent of the 3D velocity model which does not cover our whole study area. Therefore, we create a secondary seismicity catalogue for the region to the south of the 3D model. Our catalogues provide a dense, high-resolution dataset, with large spatial extent, for the interpretation of seismic processes along the western
South American Margin.

## 1 Introduction

Seismology is a fundamental tool for investigating the kinematics and dynamics of geological processes, with event detection and phase picking being the core of most seismological studies. As this is a very time-consuming task, automatic methods have been developed, such as STA/LTA triggers. However, in the past these did not achieve the accuracy of a human
seismologist (Küperkoch et al., 2012; Bormann and Wielandt, 2013; Münchmeyer et al., 2022). Recent advances in technology, particularly in machine and deep learning methods, have allowed the development of automatic algorithms that solve this problem, achieving results comparable to those of a human in much less time (Münchmeyer et al. 2022). These

advances have led to the capacity of processing ever larger datasets, a clear advantage in seismology as denser and larger seismic networks continue to produce growing volumes of data.

The Chilean Margin, where the oceanic Nazca and Antarctic plates subduct below the continental South American plate (Kendrick et al., 2003; Breitsprecher and Thorkelson, 2009; Quiero et al. 2022), is one of the most seismically active plate boundary regions in the world and a great natural laboratory for further understanding subduction processes and the resulting seismicity. The Chilean Seismological Center (Centro Sismológico Nacional, CSN), operates a permanent network with national coverage used to routinely identify and locate events with the purpose of seismic monitoring and information delivery

to authorities and the public. Continuous waveform data since the year 2012 is available; however, the station coverage has been better since 2017, when the network was fully deployed (Potin et al., 2025). The Chilean volcanology observatory (Observatorio Volcanológico de los Andes del Sur, OVDAS), also operates a permanent seismic network, focused on volcano monitoring, and has stations on 45 active volcanoes along the Andes. Due to the different missions of both agencies, CSN stations are located mostly along the coast, closer to the megathrust, or near important cities. On the other hand, OVDAS

stations are all located along the volcanic arc, close to the eastern border of Chile. In northern Chile, the Integrated Plate Boundary Observatory Chile (IPOC), a multiparametric network operated by a consortium of European-Chilean institutions led by GFZ-Potsdam (Germany), has provided abundant and accurate earthquake locations for over 15 years, resulting in a dense seismicity catalogue (Sippl et al. 2023), but limited only to a small portion of the margin. Other regional seismic catalogues are available from international networks, but due to sparse station coverage, these have high completeness

magnitudes and location errors. Complementary to these permanent networks, many local temporary deployments of seismic arrays have been conducted throughout Chile (e.g. Farias et al., 2010; Marot et al., 2013; Sielfeld et al 2019; Pasten-Araya et al., 2022; González-Vidal et al., 2023). Put together, there are abundant seismic networks, permanent and temporary, along the Chilean Margin. Coupled with the large extent of the region and its high seismic productivity, this results in a large amount of data that is hard to process with traditional methods. For this reason, there is no large-scale, dense and high-quality catalogue

primarily aimed at scientific research. Until recently, the best available catalogue was that of the CSN. While it is based on the dense CSN and other complementary networks (networks C, Universidad de Chile, 1991; C1, Universidad de Chile 2012; IU, USGS, 1988; G, IPGP y EOST, 1982 and CX, GFZ, 2006 in Fig. 1) and is in fact quite complete for a regional catalogue, due to the CSN's mission of locating events of magnitudes 3 or higher, it misses large parts of the abundant microseismicity at smaller magnitudes that can be relevant for the study of seismic and tectonic processes. Furthermore, CSN's detection and

location philosophy has changed over time, resulting in some temporal inconsistencies (Potin et al., 2025). In a recent publication, Potin et al. (2025) improve upon this catalogue by relocating events in a new 3D velocity model. However, their catalogue is limited to the events present in the original CSN catalogue. Modern automated approaches offer new possibilities to handle data processing in this kind of setting, allowing us to take better advantage of the available raw data.

In this study, we pair modern machine learning methods with traditional location and relocation techniques to process 5 years

of data (2017-2021) from different regional networks, including some interesting earthquake sequences as: Valparaíso 2017 (Mw 6.7); Coquimbo 2019 (Mw 6.7) and Atacama 2020 (Mw 6.9). We choose a short time window and specific networks

rather than including all available data to keep network/station uptime as constant in time as possible. We create a workflow to handle the large data volume and geographical extent of the study area, obtaining a high-resolution catalogue with >600.000 earthquakes, more than 100.000 events per year, magnitudes as low as Ml 0.3 and an estimated completeness magnitude of

Ml ~2.5. We classify events in this catalogue according to their distances from the trench, slab and Moho and thus mark them as belonging to different tectonic entities, creating a powerful tool to assess temporal and spatial changes in seismicity along the margin, evaluate features of the megathrust and study crustal, interplate and intraplate seismicity separately.

## 2 Data and method

We process 5 years of waveform data from permanent CSN (networks C, Universidad de Chile, 1991 and C1, Universidad de

Chile, 2012), IPOC (network CX, GFZ, 2006) and OVDAS (network VC, Sernageomin, 2015, from which 1 station per monitored volcano was made available to us) stations, for the years 2017 to 2021, as well as some stations from other networks, scattered throughout Chile and Argentina (networks IQ, Cesca et al., 2009; IU, USGS, 1988; G, IPGP y EOST, 1982; GE, GEOFON, 1993; GT, USGS, 1993 and WA, UNSJ, 1958) for the same time period (Fig. 1a). To improve station coverage at the southern end of our study area, where CSN stations are scarce and no other permanent networks are available, we also

included stations from the temporary 1P network (Wiens & Magnani, 2018, Fig. 1a), operated by the University of Washington, which have data for about half of our study period, between 2019 and 2021 (Fig. 1b).

Although there are many other temporary deployments, we purposely chose not to include all available data, selecting only networks that were deployed for most of the study period. The objective of this decision is to produce a catalog that is as constant in time as possible (Fig. 1b). Then, the catalogue's magnitude of completeness should be approximately consistent

in time and all temporal changes should be due to the behavior of the margin itself and not artifacts of network changes, allowing for the interpretation of the temporal evolution of seismicity. Otherwise, it would be difficult to distinguish whether a particular temporal feature of seismicity was intrinsic to the seismicity or an effect of a particular station or network going on- or offline.

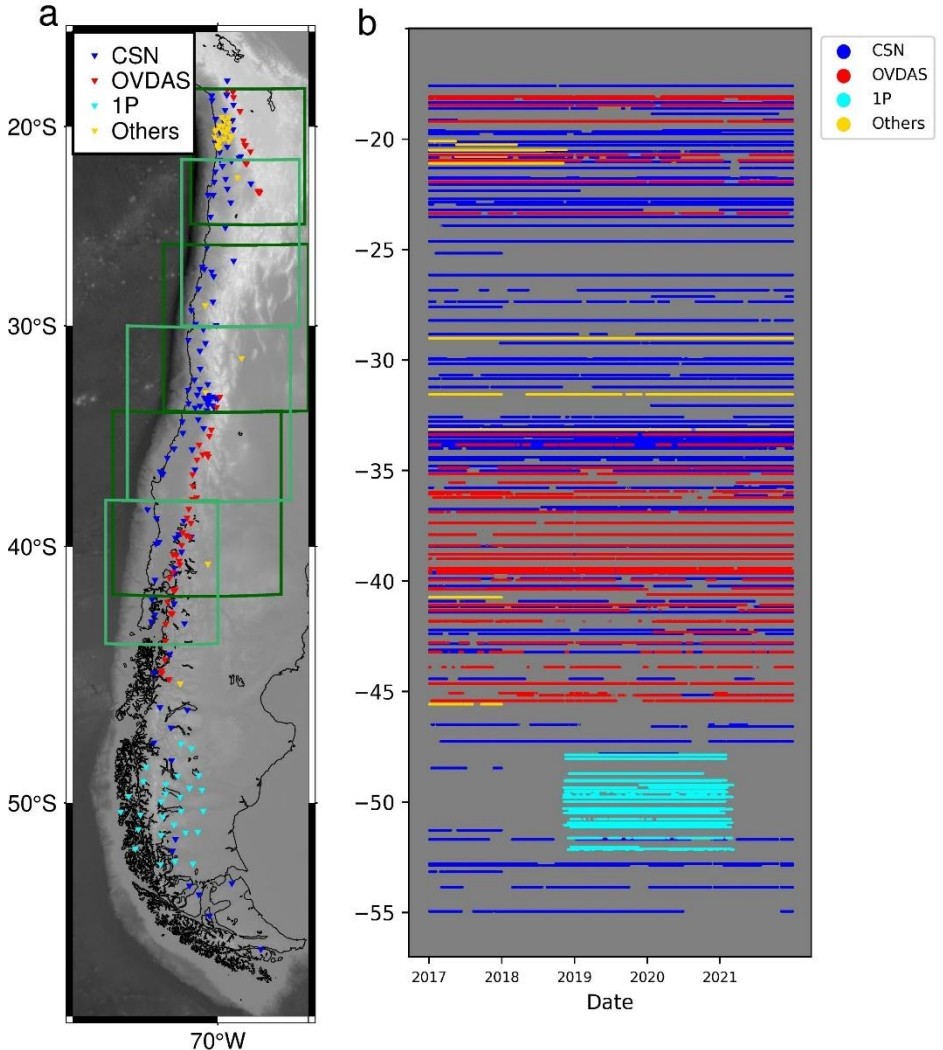

**Figure 1: a) Station distribution map and b) General overview of station availability over time. The x axis shows date, between 2017-01-01 and 2021-12-31, and the Y axis shows stations sorted by latitude and color-coded with blue for all networks used by the CSN to build their catalogue (C, Universidad de Chile, 1991; C1, Universidad de Chile, 2012; CX, GFZ, 2006; IU, USGS, 1988 and G, IPGP and EOST, 1982), red for the OVDAS network (VC, Sernageomin, 2012), cyan for 1P (Wiens & Magnani, 2018) and yellow for all others. Dark and light green rectangles on a) indicate segments of the main catalogue. Thin black lines represent political boundaries. Figures S1-S20 in the supplementary material provide a more complete overview of station availability in time.**

To build our catalogue, we created a workflow using an automatic deep-learning picker that detects seismic phases from recorded waveforms, an associator that relates detected picks to individual events, a non-linear location algorithm and a double-difference relocator.

We employ Earthquake Transformer (Mousavi et al. 2020) as a phase picker. Rather than train a model for our study area, we chose a version pretrained with the INSTANCE catalogue (Michellini et al., 2021), available in the SeisBench library (Woollam et al. 2022). We tested different pretrained models and trained a model with picks from the IPOC dataset, obtaining good results with the INSTANCE pretrained model. Furthermore, Münchmeyer et al. (2022) systematically compared the in-domain (training dataset is from the same region where model is applied) and cross-domain (training dataset is from a different region as that where the model is applied) performance of machine-learning pickers included in SeisBench and found that, overall, models trained on INSTANCE perform best. We employ a probability threshold of 0.08 for P-picks and 0.1 for S-picks.

The resulting picks must then be associated to link them to earthquakes. Based on work by Puente et al. (2025), we chose PyOcto (Münchmeyer, 2024) for phase association, initially using a 1D velocity model proposed for the South-Central Andean region (Bohm et al., 2002). A common issue with grid-based associators like PyOcto is the handling of Earth's curvature for large geographical areas as our study region, as they normally use a cartesian coordinate system (e.g. Münchmeyer, 2024; Zhang et al., 2019). To circumvent this problem, we divided our study area into overlapping latitudinal segments for phase association, so that each segment is small enough to be processed through a local cartesian system (Fig. 1a). Within each segment, we defined an event as having at least 8 total picks, with at least 4 stations having both P and S picks.

After association, events for each segment were relocated using NonLinLoc (NLL, Lomax et al., 2000) with a 3D velocity model proposed by Potin et al. (2025). This is the existing 3D velocity model with the largest extent for the study area (18º-43ºS), although it does not cover the southernmost segment of the Chilean margin. Therefore, we chose to create a second dataset, with events that fall outside of the 3D model between 43°S and 55°S. For this subset, we relocated events using the 1D velocity model for the Austral Andes by Ammirati et al. (2024). Therefore, our catalogue is divided into two datasets: A main catalogue relocated with the 3D velocity model by Potin et al. (2025) and its corresponding picks and a secondary catalogue, located with the 1D velocity model by Ammirati et al. (2024) and its corresponding picks (Riedel-Hornig et al., 2025). For consistency, we extended the secondary catalogue up to 39°S, resulting in a 4° overlap between the catalogues, and both were processed with the same workflow.

NonLinLoc probabilistically locates events by defining a probability density function (PDF) in 3D for the hypocenter location (Lomax and Savvaidis, 2021). Application to all events identified in the phase association step provides a preliminary seismic catalogue. However, due to segment overlap, this catalogue contains abundant duplicate events. We identified duplicates in the overlapping regions as events that are separated by less than 10 seconds in origin time, that have hypocenters located within 0.5° of each other in latitude and longitude and 50 km or less difference in depth, and that have shared picks. Due to the segment overlap, it is guaranteed that duplicate events will share at least a few picks, for stations falling within the overlapping region, therefore providing a consistent way to distinguish them from earthquakes that may nucleate very close together in space and time. Segment overlap also provides a consistent way to avoid location artifacts that may arise when locating events at the edges of the velocity model. Thus, all located events have stations both to the north and south. Should a hypocenter be located at the edge of one segment, then there will be a duplicate event within the adjacent segment that is

not at the edge, but further inside of said segment. We keep the event closest to the center of each segment and discard the ones at the edges.

From duplicate event pairs, we discarded the event with higher location errors or closest to the edge of a segment and, when necessary, added missing picks (picks from stations in the adjacent segment originally assigned to the discarded event) to the event with the lower location errors. We also removed picks that had residuals that exceeded 2 standard deviations of the

event's mean residual or with an absolute residual >2 s, as in Sippl et al. (2018). Since some picks from the discarded events were added to the remaining ones and some picks were removed, we relocated events in NLL once again. Absolute location errors are taken from these locations and represent the size of the ellipsoids that contain 68% of the PDF (Lomax and Savvaidis, 2021).

Finally, to sharpen feature geometries, we performed double-difference relocation with HypoDD (Waldhauser and Ellsworth,

2000). In this process, cross-correlation times were not used. Since this software has trouble handling large amounts of data, we followed the approach of Sippl et al. (2018) and subdivided the study area into new overlapping segments, of roughly 30.000 events each (close to the limit of HypoDD's processing capacity). Events within each segment were relocated, with some being relocated twice, due to segment overlap, in which case relocation coordinates were averaged between segments. This step was only performed for the main catalogue, as lower event density in the southern end of the study area did not

allow for proper clustering necessary for relative relocation by HypoDD in the secondary catalogue.

Local magnitudes for both catalogues were calculated using the maximum amplitudes on the horizontal components after Hutton and Boore (1987), using the Bormann (2012) attenuation function, as recommended by the IASPEI. However, response information was not available to us for network VC. Therefore, for events that are located mostly with VC station picks, we were unable to calculate magnitudes.

Finally, we classified events within the main catalogue according to their distance to the trench, slab surface and continental Moho. Events west of the trench, plus a 0.25° error margin, are classified as "Outer rise" Here, we consider a wide error margin as all outer rise events are outside of the network and therefore have higher location errors than other earthquakes. Events within a set margin above or below the Slab2 slab surface model (Hayes et al., 2018), down to a 60 km depth, are classified as "Interplate". This margin was obtained by dividing the area into 0.25° by 0.25° cells, and calculating the average

depth error for events within said cell down to a minimum of 2.5 km and up to a maximum of 10 km. We choose to calculate the margin in this way to account for higher errors towards the outside of the network and discrepancies with the Slab2 model. The 60 km depth limit is set because in some regions of our study area, the resulting seismic catalogue displays a "triple seismic zone", where the upper and lower seismicity planes of a double-seismic zone (DSZ) are clearly distinguishable from a shallower plane that follows the plate interface, down to ~60 km depth. We note, however, that this may result in some

intraslab events being classified as plate interface events, as the coupled portion of the plate interface does not terminate at a constant depth, but this rather depends on the thermal structure. Analyzing thermal structure to obtain a better classification in this sense is beyond the scope of this study. Events below the slab surface not fitting this criterion are classified as "Intraslab". Events shallower than the continental Moho are classified as "Crustal". We use the Moho from the model of

Tassara and Echaurren (2012). Events that do not fit any of these criteria are labeled as "Not Classified". This results in a simple classification of events, that may be useful for regional interpretation of seismicity. However, due to location errors uncertainties in the Slab2 and Moho models, the narrow width of the subduction channel and the proximity of some intraslab and crustal events to the plate interface, some events may be mislabeled. We especially highlight possible uncertainties induced by discrepancies with Slab2 as events in our and previous (e.g. González-Vidal et al., 2023, Sippl et al., 2023 and Hernández-Soto et al., 2024) catalogues show a deviation from this model.

Earthquakes in the secondary catalogue were not classified as slab and Moho models do not extend as far south as the catalogue.

## 3 Results

### 3.1 Main seismicity catalogue

Here we only describe results for events located with the 3D velocity model by Potin et al. (2025) between 18ºS and 43ºS. The aforementioned process produced over 40 million picks, from which 6,966,042 P and 5,302,574 S picks were used for association for a total of 621,917 events (Riedel-Hornig et al., 2025 and Fig.2a). From this, 5,099 events were classified as outer rise, 66,284 as interplate, 65,136 as crustal and 459,208 as intraslab events. It must be noted however, that this may include misclassified events due to local inaccuracies of the Slab2 model. Intraslab earthquakes are the most abundant, with over 70% of all events falling into this category. Figure 2 b-d displays event density for crustal, interplate and intraslab events, demonstrating that this last group of events is the most widely distributed and has the highest densities. Two particularly active segments (~20°-24° S and ~30°-34° S) can be distinguished in the maps as presenting high seismicity rates throughout the different event categories (although this is most pronounced for intraslab events). These segments coincide with areas of higher station coverage (Figure 1) and thus events with lower magnitudes are more easily detected here, partially explaining that they appear to be more active than adjacent areas. However, this feature remains even when filtering the catalogue by Mc or well above it.

Cross-sections in Fig. 3 further demonstrate the heterogeneity in hypocenter distribution. These cross-sections display the event classification, but also illustrate the resolution of our catalogue, displaying features such as a clear DSZ at least in cross-sections 1-3, 5 and 7, and less clearly in others. Where event density is higher, a "triple seismic zone" can be seen, with the upper and lower seismicity planes of a DSZ distinguishable from a third plane at the plate interface (cross-section 2 and possibly 3). Some of the observed features, for example strangely scattered intraslab seismicity at depth below 100 km in cross-sections 3 and 7 are partially due to profile width (30 km) in an area with fast latitudinal changes in slab geometry but also influenced by higher location errors in this area, where events that do not get picks form the stations at the eastern end of the network may be mislocated, something we will address in the discussion. It must also be noted that events inside the network are less scattered than those to the west and east, which in some areas (e.g. very deep outer rise seismicity in cross-

section 1) may have high scattering. This is expected as location error increases outside of the network, which is reflected by our location error estimations (Fig. 4).

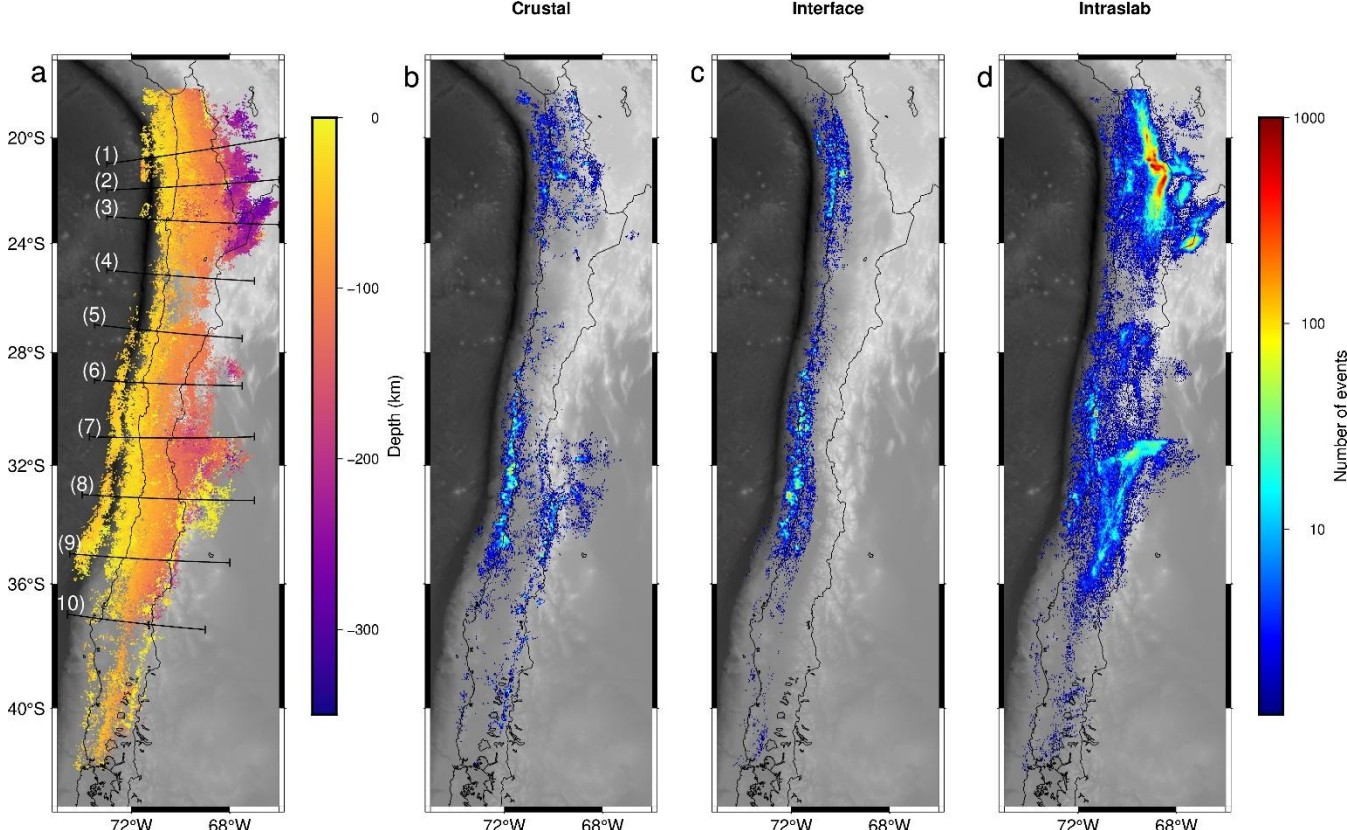

**Figure 2: a) Map of all hypocenters in our catalogue located with the 3D velocity model from Potin et al. (2025), plus location of the**
**cross-sections shown in Fig. 3. b-d) Density maps for crustal, interplate and intraslab events, respectively, calculated as the number of events within each cell (2 by 2 arc minutes, or 0.03°). Thin black lines represent the coastline.**

Figure 4 displays absolute location errors taken from NLL output, which average 4.8 km in longitudinal, 2.2 km in latitudinal and 6.1 km in vertical direction. Due to the network geometry, latitudinal errors (Fig. 4b) are lowest. Longitudinal and depth errors (Fig. 4 b-c) are highest outside the network, towards the west and east, which is where the strongest event scattering is
observed in the cross-sections as stated above. A good example of this are outer rise events, all of which are located outside the network and have high location errors, especially in depth. Very deep outer rise events, such as those in cross-section 1, do not have a well constrained depth. Onshore and within Chile, where most stations are located, absolute errors are generally lower than 5 km.

    We calculated the magnitude of completeness (Mc) of our catalogue through the maximum curvature method (Mignan and
Woessner, 2012), obtaining a global Mc of 2, much lower than any previous regional catalogue for our study area. However,

spatial network heterogeneity must be considered. Particularly, from ~24°S to 27° S and from ~41°S to 43° S, station coverage is far less dense than in other areas, resulting in lower event densities as shown in Fig. 2b-d and cross-sections 4-5 and 10. Furthermore, in those two regions station distribution follows a thin N-S line, making earthquake locations even more error-prone. Therefore, an overall Mc for the entire catalogue may not be representative, and its spatial distribution should provide better information. Local estimates of Mc confirm this assertion, showing areas with Mc lower than 1.5 for regions of high station coverage (near Santiago for example) and as high as 4 on the outskirts of the network (Fig. 5). Within Chile, the areas with highest Mc are around 2.5. We therefore believe that Mc = 2.5 should be a safe and representative average value for our catalogue.

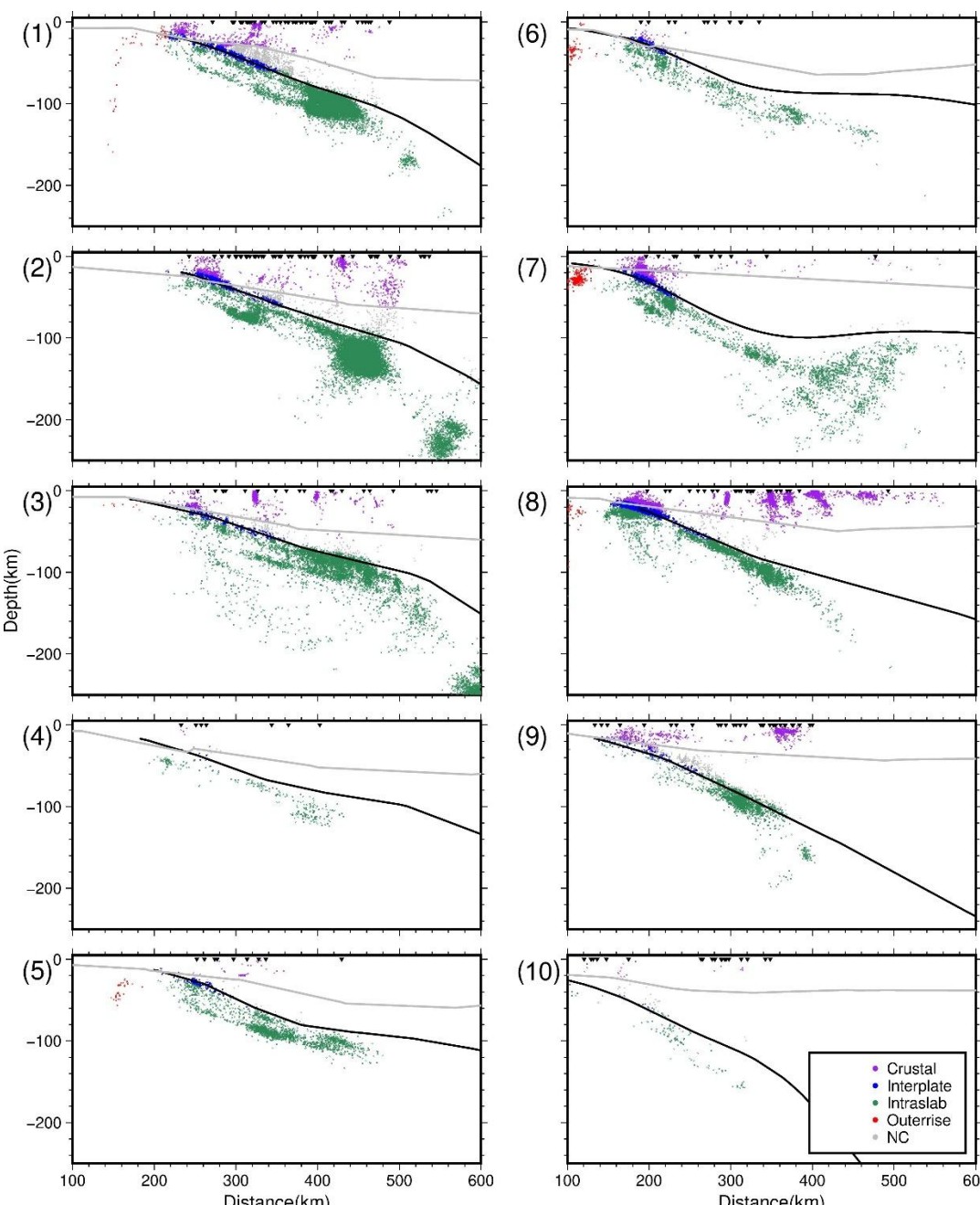

**Figure 3: Cross-sections highlighting event classification. Each section includes a 30 km wide swath of seismicity (i.e. 15 km north to 15 km south of the cross-section line as shown in Fig. 2). Black lines indicate the slab surface after Slab2 (Hayes et al., 2018) and gray lines the continental Moho (Tassara and Echaurren, 2012). Black triangles represent stations within 200 km north or south of each cross-section. Distance is calculated along each cross-section, starting at the beginning of the cross-section as mapped in Fig. 2.  NC= not classified.**

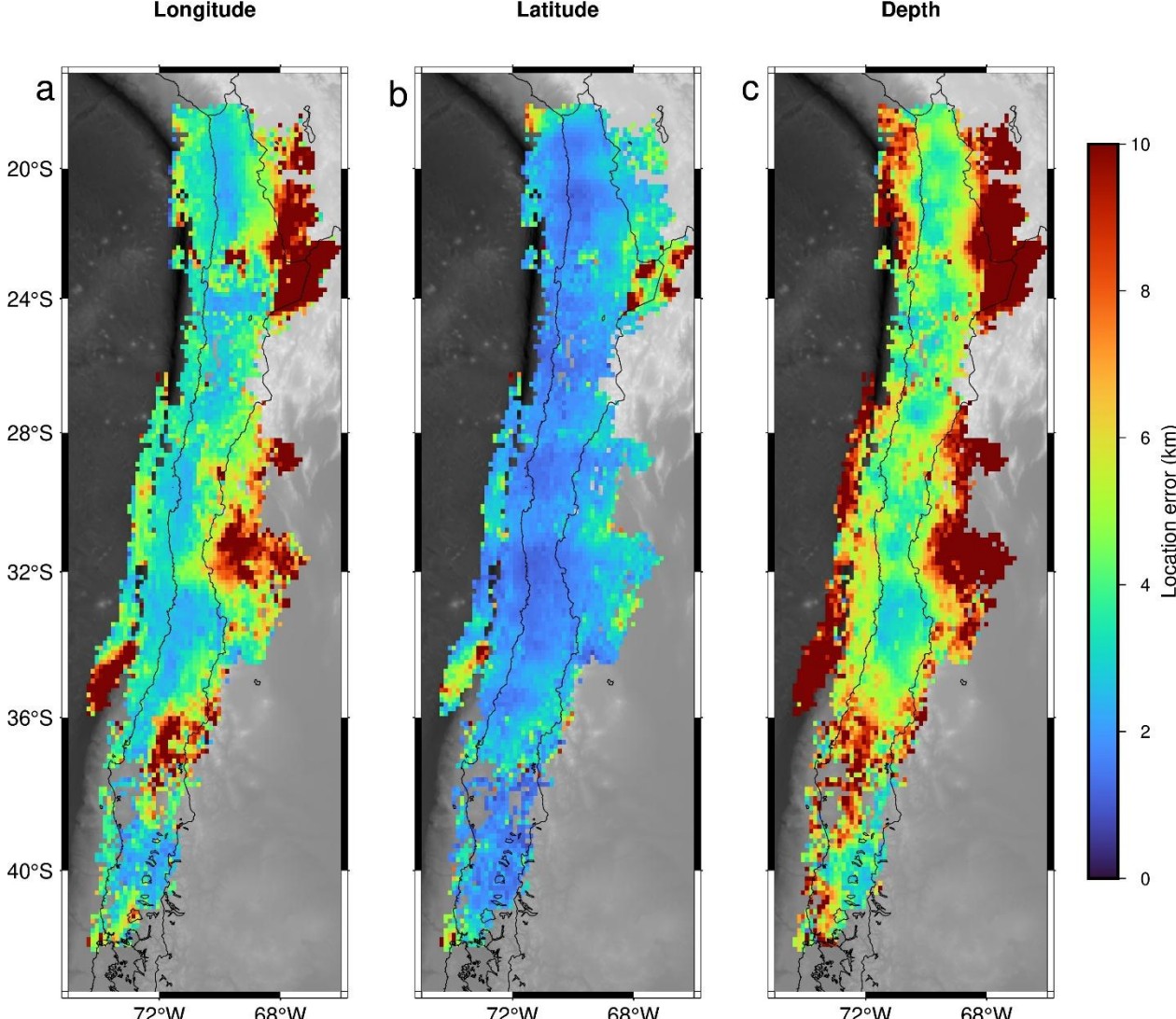

**Figure 4: Absolute event location error estimates, calculated as the average error of all events within each cell. a) Longitudinal error, b) Latitudinal error, c) Depth error. Dark red areas indicate errors of 10 km or more. Thin black lines represent the coastline.**

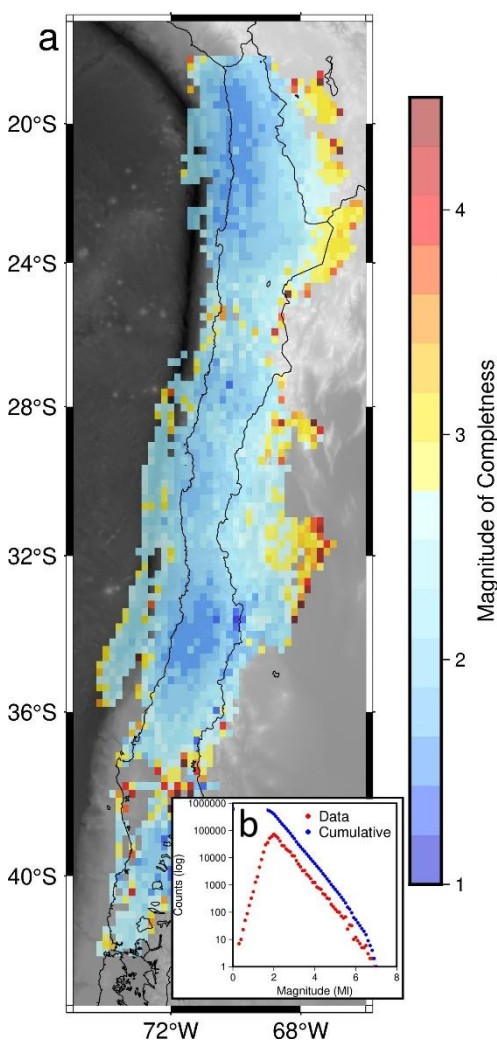

**Figure 5: Magnitude of completeness of our catalogue. a) Mc map and b) Magnitude distribution for the whole catalogue.**

## 3.2 Secondary seismicity catalogue

The secondary catalogue consists of 112,134 P and 72,072 S associated picks (Riedel-Hornig et al*., 2025), corresponding to 12,933 events (Riedel-Hornig et al*., 2025 and Fig. 2a). Event density is considerably lower than further north, in the area covered by the main catalogue. The area with the lowest seismic productivity is around 44°S to 50° S (Fig. 6a), where the Chile Rise subducts and the oceanic plates (Nazca Plate to the north and Antarctic Plate to the south) are youngest. While events here are not classified, given their depth and location (Fig. 6a-c) most of them are likely to be crustal events.

Cross-section b in Fig. 6 displays the Nazca slab and crustal events, with no clear features distinguishable within each of these groups, which may be a direct effect of lower seismic productivity in the region and slightly higher location errors (Fig. 6e-g) resulting from the usage of a 1D velocity model only. In cross-section c, most events seem to be crustal, with on average

deeper hypocenters towards the edges of the cross-section and  shallower ones in the middle, where the Southern Patagonian Icefield (SPI) is located. Overall Mc is higher than in the main catalogue, estimated at 3 (Fig. 6d).

Event density is considerably lower in the secondary than in the main catalogue. Even if we select only those events with magnitudes above 3 (i.e. higher than the Mc of both catalogues), the region covered by the secondary catalogue shows a much lower event density. Moreover, within this region around ~46°S, at the triple junction between the Nazca, Antarctic and South American plates, there are almost no events. As mentioned above, this is possibly related to the subduction of the Nazca-Antarctic spreading center, although other interpretations are also possible. While this might also be influenced by a considerably lower station density and shorter network runtime (1P Network, Wiens & Magnani, 2018, Fig. 1) in the area, this alone should not produce the observed decrease in seismicity as it is a feature that remains even when filtering events by Mc.

245

250

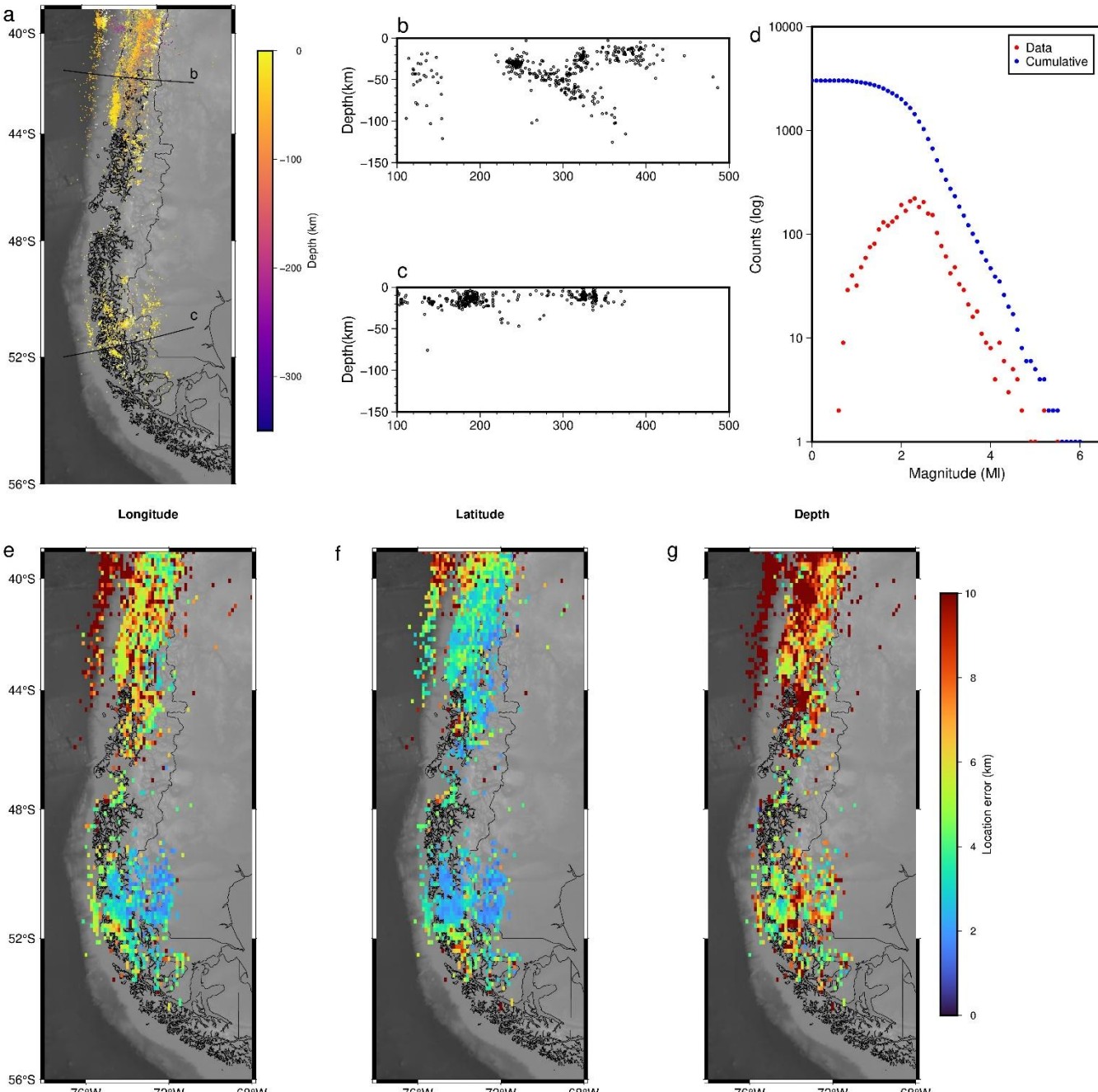

**Figure 6: a)** Map of the secondary seismicity catalogue. Dots indicate hypocenters, black lines indicate cross sections b and c. **b-c)** Cross-sections as mapped in a. Each one includes events from 30 km north to 30 km south of the cross-section. **d)** Magnitude distribution of events in the secondary catalogue. **e-g)** Absolute longitudinal, latitudinal and depth location errors, calculated as the average error for all events within each cell.

## 3.3 Continuity in time

Due to our data selection as described in Sect. 2, the resulting catalogue should be largely homogeneous in time (Fig. 6), yet heterogeneous in space. This allows for the spatio-temporal assessment of seismicity, where most changes should be a characteristic of seismicity and not of the network. The only exception to this continuity is network 1P, which was active only for about half of our study period (Fig. 1b). We chose to include it nonetheless, due to a large lack of stations along the southernmost and austral Andes. This must be considered for any interpretation of our secondary catalogue.

While station outages and data gaps could induce changes in seismicity over time, which we tried to avoid, large earthquakes or other short-term manifestations of tectonic processes can also produce such changes. Within our study period, no large megathrust earthquakes occurred along the Chilean margin. This is reflected in Fig. 7 by the continuity of event density in time, specifically on the plate interface, and by the stability of magnitudes over time (Supplementary Fig. S21). Seismicity can be distinguished in Fig. 7b, which corresponds to persistent clusters and swarms offshore Valparaiso, the 2017 6.7 Mw interplate event (Ruiz et al., 2017) and the aftershocks of the Mw 6.2 Rio Loa crustal earthquake (Tassara et al., 2022; Sippl et al., 2023). While there are many M~6.0-6.9 intraslab events, these do not seem to produce clearly distinguishable aftershock activity.

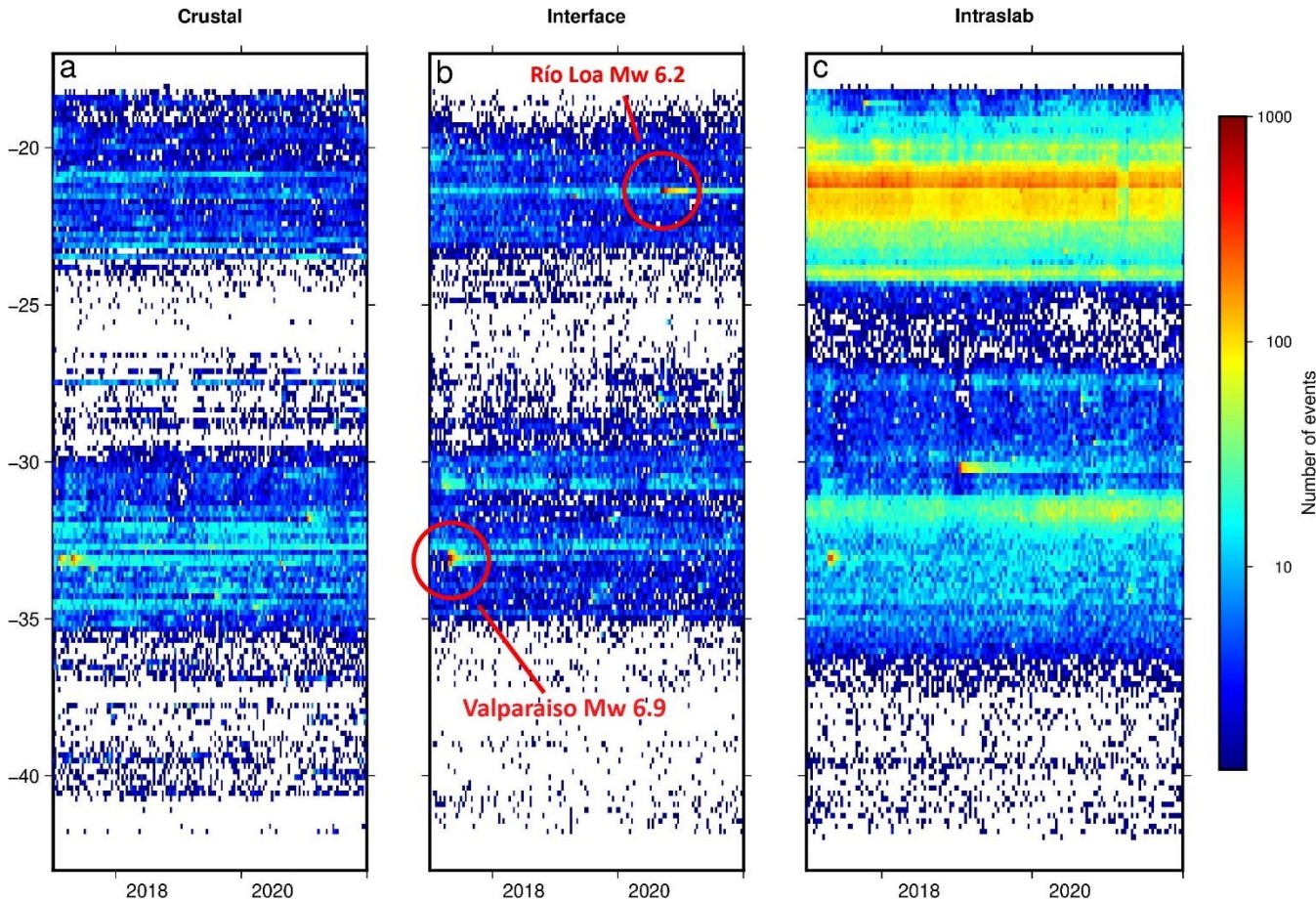

**Figure 7: Number of events per cell (of 12 days by 0.17°) in the main catalogue over time. Y-axis indicates latitude and X-axis time. a) Crustal events over time, b) Interplate events over time, c) Intraslab events over time. Dark red areas contain 1000 or more events.**

## 4 Discussion

### 4.1 Catalogue features and comparisons with previous studies

As described in the previous sections, there are abundant seismic networks in Chile and neighboring countries. While new technologies provide the possibility for faster and better phase picking and association of large datasets, the dimension of our dataset still proves to be a challenge. Seismological processing (phase detection, association, location and relocation) is not optimized for large areas, as most codes rely on a local cartesian coordinate system, where, if the study area is large, Earth's curvature produces inaccuracies. HypoDD, a standard and widely used relative relocation software, is also limited in its processing capacity and cannot relocate sets of >50,000 events, while our main catalogue features over 600.000 events. We developed a workflow that circumvents these challenges, as described in Sect. 2.

To demonstrate that this procedure yields trustworthy hypocentral locations, we resort to two different approaches. Firstly, we want to compare our results with existing catalogues. The CSN catalogue and its recent update by Potin et al (2025), likely the most complete regional catalogue in the study area, has a completeness magnitude of 3-4 over most of Chile (Potin et al., 2025), significantly higher than the estimate of 2.5 for our catalogue. This implies that the CSN catalogue has approximately 10 times fewer events per year, so that the bulk of the microseismicity that makes up our catalogue is not contained in it. This difference is partially due to the focus of CSN on events that may impact society, i.e. events with higher magnitudes that will be felt. Furthermore, due to changes in methodology and event selection as well as station coverage over time, the CSN catalogue's Mc is highly heterogeneous in time (Potin et al., 2025). Due to these shortcomings, it does not make sense to compare individual locations, but we only use it as a reference for large-scale features of the subduction zone, mainly the general geometry of intraslab seismicity and the location of highly active clusters. As shown in Fig. 8, general geometries agree in all the plotted cross-sections, and most crustal, plate interface and intraslab clusters coincide between both catalogues, with our 5-year catalogue consistently having higher event numbers than Potin et al's (2025) version of the CSN catalogue, which covers the years 1985 to 2020.

Many known local seismicity features have so far been obtained from the CSN catalogue. For example, Valenzuela-Malebrán et al. (2021) processed data between 33° and 36°S and identified two continuously active megathrust seismicity clusters, the Vichuquen and Navidad Clusters, as well as the seismic sequence of the 2017 Valparaiso earthquake. Figure 9d shows a comparison between the event density in that area for the CSN and our catalogue. All three clusters (Navidad, Vichuquen and the Valparaiso Sequence) display the same spatial behavior between both catalogues, with the higher event density in our catalog. However, some finer-scale features that require high event density and resolution to identify, such as the exact geometry and along-strike variations of the DSZ or local clusters within the upper plate crust, are more clearly present in our catalogue since they do not show up clearly in the CSN catalogue.

To overcome the completeness limitations of the CSN catalogue, we turn to two examples of available smaller-scale but more complete seismicity catalogs along the Chilean margin. Using 15 years of data from the IPOC network, Sippl et al. (2023), identified an anomalous geometry of intraslab earthquakes in northern Chile, around ~21°S. There, a dense, highly active intermediate depth seismicity cluster displays a peculiar geometry, with a sudden E-W offset in seismicity. Like the IPOC catalogue shows, north of 21°S, the intermediate-depth earthquakes of our catalogue are mostly oriented N-S, then there is a gap in seismicity and the cluster is displaced eastward, now following a WNW-ESE trend, only to stop at another gap, after which it resumes a ~N-S trend (Fig. 9d, right).

In our second example, we leverage the seismicity catalog of González-Vidal et al. (2023), which came from a local network at 23-30°S that was deployed for 15 months starting on November 2020. In their catalogue, they can identify a DSZ between these latitudes, which had previously not been seen in regional catalogues (Fig. 9c). A more recently published study by Münchmeyer et al. (2025) used a larger dataset and obtained a significantly larger catalogue, but their observation of a DSZ in this area is largely equivalent. While we did not use these temporary stations for our study due to their short time coverage, we are still able to retrieve this feature (Fig. 9c).

As shown in Fig. 9b-d, we can retrieve the abovementioned features (Central Chile clusters, anomalous subduction zone in northern Chile, DSZ between 23°-30°S) that have previously been identified in different catalogues. This demonstrates the advantage of our catalogue, which is much more complete than previous regional or global catalogues (eg. CSN, NEIC, ISC), but offers a larger spatial extent than local catalogues based on temporary deployments. While on a local scale, a catalogue such as that of González-Vidal et al. (2023) may have higher resolution than ours due to the usage of dense but temporary networks, comparing between different catalogues, created with different workflows in different areas is not straightforward, hampering the joint interpretation of regional features that may be present in different catalogues. For example, DSZs have been identified in Chile through the IPOC network in northern Chile and through temporary local networks around 32°S (Marot et al., 2013) and between 23°-30°S (González-Vidal et al., 2023 and Münchmeyer et al., 2025). With previous data, it was unclear whether a DSZ is a continuous feature along the Chilean margin, or whether these three observations mark three separate, spatially limited occurrences of DSZs. With our data, we can state that the DSZ appears to be present along most of the margin between latitudes 18°S to 35°S, allowing for the interpretation of factors influencing regional changes in its geometry and seismic productivity which opens the possibility for further research.

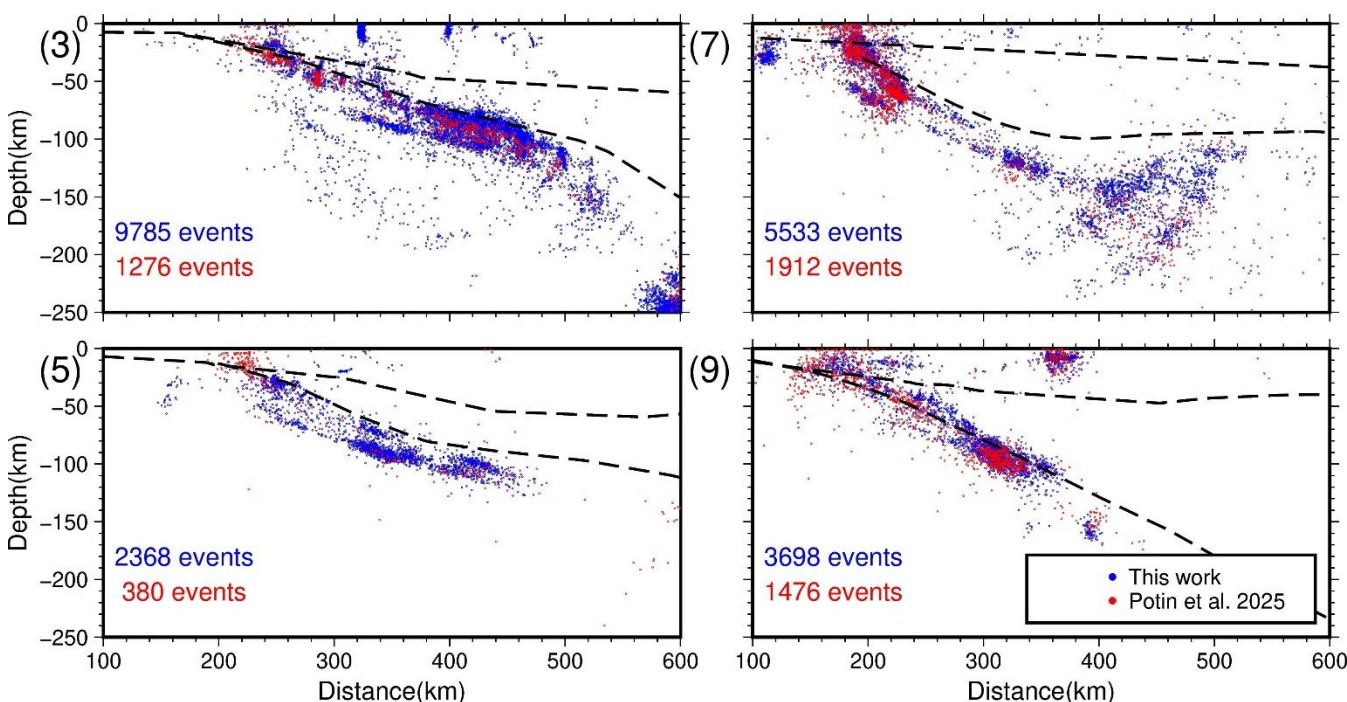

**Figure 8: Cross-section 3, 5, 7 and 9 from Fig. 2 but plotting our seismicity catalogue (2017-2021) and the updated version of the CSN catalogue by Potin et al. (2025; 1982-2020).**

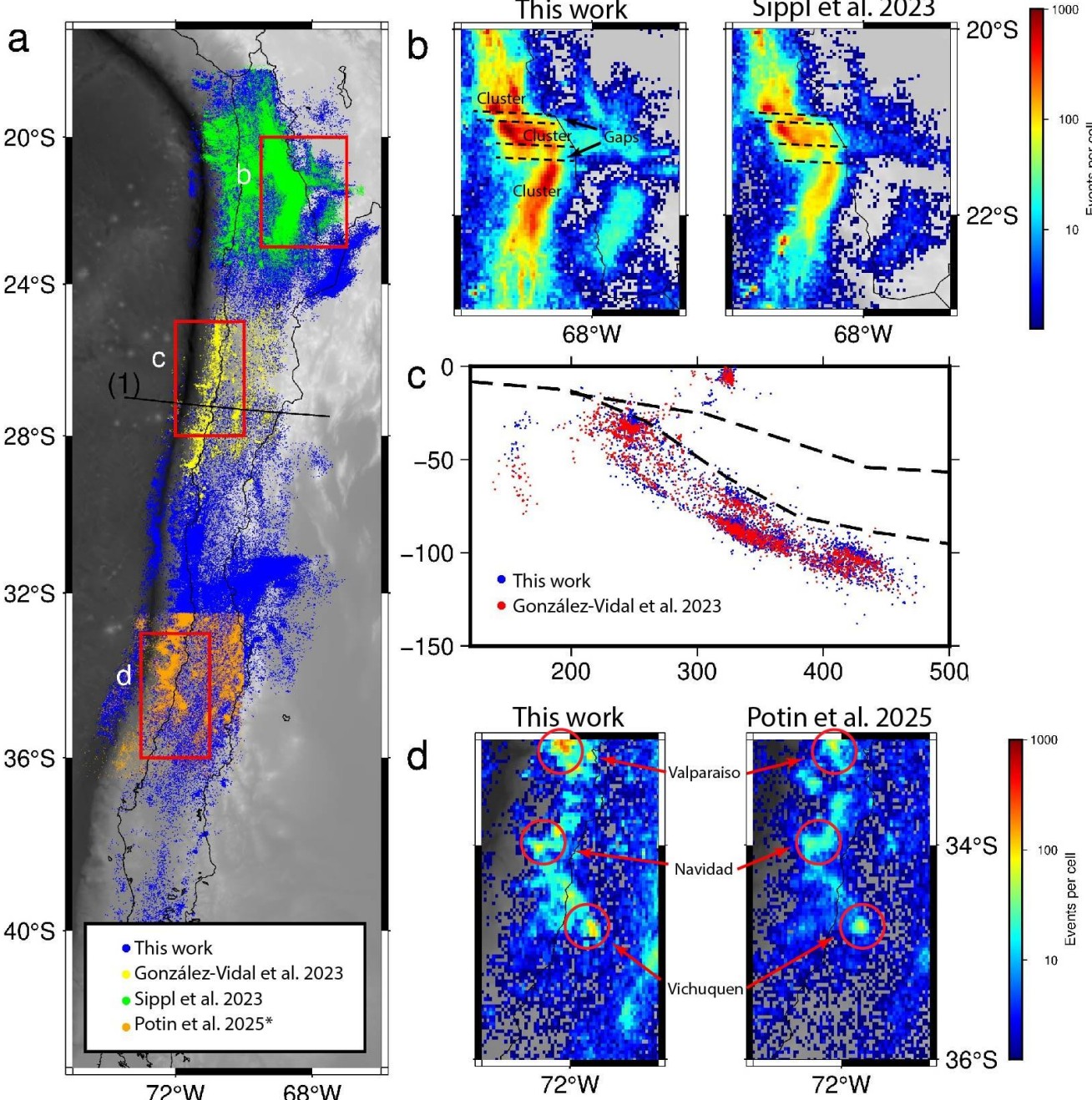

**Figure 9: Maps comparing our seismicity catalogue to previous regional and local ones. a) General map of our as well as previous seismicity catalogues. b) Maps comparing the event density of our catalogue (left) to the IPOC catalogue (right: Sippl et al., 2023). c) Cross section (1), comparing our catalogue and that of González-Vidal et al. (2023). d) Maps comparing the event density for our (left) and Potin et al. (2025, right) catalogue, and displaying the Valparaiso, Vichuquen and Navidad seismicity clusters. *Catalogue is clipped to the study area of Valenzuela-Malebrán et al., 2021.**

South of our primary seismicity catalogue, in the region of our secondary seismicity catalogue, there are few seismic studies. Adaros et al*.*, (2003) created a velocity model, which was later refined by Guzmán (2020) and then by Ammirati et al., (2024), this last update being the one we employ here. They process data form network 1P, using the LASSIE method from pyrocko to obtain 2932 events (Ammirati et al*.,* 2024) and observe that almost all seismicity in the area is crustal, being deeper around the South Patagonian Icefield (SPI) and shallower beneath. They interpret this as representing a relationship between crustal seismicity and glacial unloading and a possible locking of the plate interface. While we do not venture into interpreting the meaning of the spatial distribution of crustal seismicity and lack of interface events in this region, our results coincide with those of Ammirati et al., (2024); we observe shallower seismicity below the SPI and deeper around it and almost all seismicity in the area appears to be crustal with almost no plate interface events, in strong contrast to the rest of the Chilean margin.

## 4.2 Discrepancies with slab models and quality of deep intraslab locations

The upper termination of intraslab earthquakes in our catalogue is generally deeper than the slab upper surface according to the Slab2 model (Hayes et al*.*, 2018), especially at depths below 100 km, as can be seen in Fig. 3. The same pattern has been seen in the CSN (Fig. 8) or IPOC (Sippl et al*.*, 2023) catalogues. Previous authors already suggested that slab dip, especially in northern Chile, may be steeper than shown in Slab2 or Slab1 (e.g., Tassara & Echaurren, 2012; Sippl et al., 2018; Hernandéz-Soto et al*.*, 2024). This makes it difficult to interpret whether intraslab events occur within the oceanic mantle, oceanic crust or close to the slab surface. It is not in the scope of this study to answer this question, but whichever the case, events should not systematically deviate from the slab surface as is the case here, which suggests that currently used slab models may be in error. However, most of the aforementioned studies, just as our work, are based on data from seismic stations located mainly within Chile, utilizing only few available stations from international networks in Argentina. Thus, hypocenters located to the east and outside of the networks in Chile are not well constrained, especially in terms of hypocentral depth. Figure 4 depicts this issue, where the easternmost hypocenters have highest latitudinal and depth errors. To test whether the observed deviation from Slab2 may be a network artifact, we obtained phase arrival times from Argentinian stations of INPRES (Instituto Nacional de Prevención Sísmica, Argentinian seismic monitoring agency) for events listed in the ISC Bulletin (ISC, 2025). We selected all INPRES events available for the year 2018 and their picks, and matched them to events in our catalogue by their origin times and hypocentral coordinates. We were able to match 2,598 events in total. We then appended the INPRES picks to our data for those events and relocated these events using the same workflow as for our original catalogue. The results are depicted in Fig. 10. When adding INPRES picks to events located east of central Chile, where the Pampean flat slab has been described (Fig. 10c, cross-section 2), events became considerably shallower and more consistent with a sub-horizontally dipping slab, while there is no clear observable change a few degrees to the north (Fig. 10c, cross-section 1).

This test likely implies that in central Chile and Argentina, hypocenters located east of the seismic network are systematically located deeper than reality and move closer to the Slab2 upper surface when considering stations further east. This is clear in Fig. 10c, cross-section 2. While on cross-section 1 this effect is not as clear, lesser event density in this region hinders accurate interpretation.

The greater inaccuracy of our hypocenters in this particular region is likely due to station distribution. As Fig. 1 depicts, the CSN network has few stations at ~30°-32°S and they are located close to the coast, the OVDAS network has no stations close to the Chile-Argentina border here since the volcanic arc is inactive and there are no international stations available, save for one in Argentina which, due to its distance to other stations, is unlikely to contribute many picks. This test highlights the importance of considering stations within Argentina when studying intermediate depth or deep intraslab events along the Andean margin.

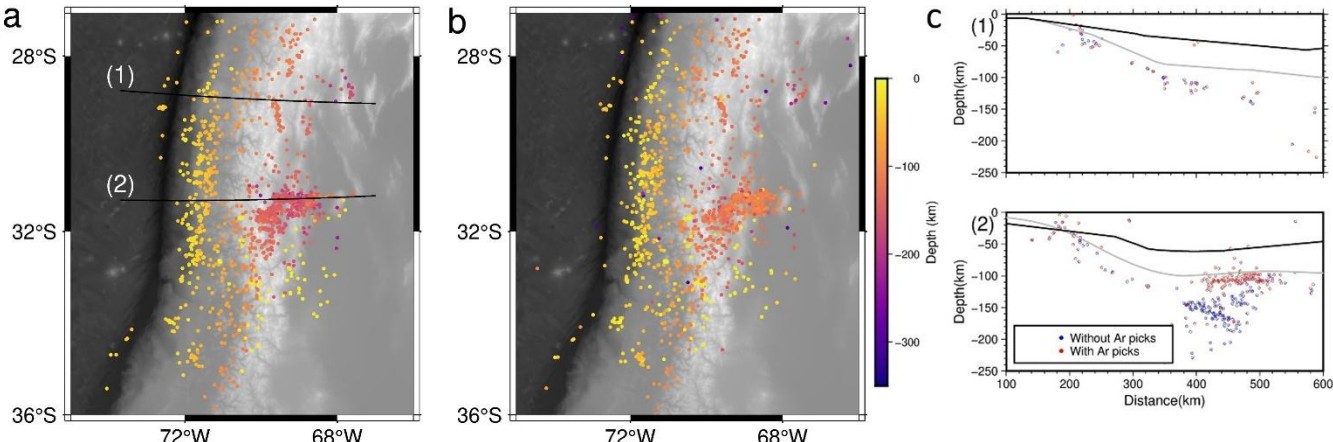

**Figure 10: a-b) Maps showing only earthquakes for which we retrieved and added INPRES picks from the ISC Bulletin a) Hypocenters with our picks only. b) Hypocenters after including INPRES picks. Black lines indicate cross-sections 1-2. 1-2) Cross-sections comparing hypocenters when including (red) and not including (blue) INPRES picks. Each cross-section contains a 30 km swath of seismicity, i.e. events 15 km to the north and south of each cross-section. Black lines indicate the continental Moho (Tassara and Echaurren et al., 2012) and gray lines the slab (Hayes et al., 2018).**

## 5 Conclusion

We integrated data from the CSN (C1, Universidad de Chile, 1991; C, Universidad de Chile, 2012), IPOC (CX, GFZ, 2006), OVDAS (VC, Sernageomin, 2012) and other (1P, Wiens & Magnani, 2018; IQ, Cesca et al., 2009; IU, USGS, 1988; G, IPGP y EOST, 1982; GE, GEOFON, 1993; GT, USGS, 1993; WA, UNSJ, 1958) seismic networks and processed 5 years of raw waveform data for events between 18°-56° S and 66°-75° W, covering the entire Chilean convergent margin. Handling the large data volume and geographical extent of the study area required the development of an automatic picking and association workflow paired with traditional location and relocation methods. This allowed us to obtain a high resolution, dense seismicity

catalogue located using a 3D velocity model northward of 43ºS as well as a smaller, secondary catalogue located with a 1D velocity model for the southernmost region.

While other regional and global catalogues are available for the study area (e.g., CSN, NEIC, ISC), they do not achieve the number of events and low magnitudes contained within our dataset. This is mainly due to different objectives, but also due to our inclusion of different networks. For example, it would be impossible for a global catalogue such as NEIC to have the station density required to detect microseismicity. The CSN catalogue, which draws upon abundant stations in our area of interest, has the objective of locating events that might have been felt by the public and providing reliable seismological information, thus is geared towards mainly detecting events with magnitudes higher than 3. This implies that their dataset misses many smaller events that might also be relevant for the study of subduction processes.

Our choice of data and workflow are aimed at providing catalogues that are as continuous in both space and time as possible, allowing for a consistent interpretation of regional natural variations in seismicity. We are able to accurately retrieve features of the megathrust and inside both converging plates that have already been identified through independent, separate, local seismic networks. We identify these features with higher resolution than previous regional catalogues. On the other hand, while it would be hard to obtain higher local resolution than with dense, temporary networks, our catalogue has a larger spatial extent. The main advantage of our result thus is the pairing of these features: optimized temporal and spatial consistency, high resolution with low Mc and large spatial extent. Therefore, we believe that our catalogue provides a powerful tool for the study of seismicity and subduction processes along the Chilean margin.

**Data availability**

Access links for the data used in this paper are provided in Table 1. This includes the seismic catalogues and phase picks, provided with the dataset publication (Riedel-Hornig et al., 2025) under Creative Commons Attribution 4.0 International license and the compilation of seismic networks used to build the catalogue.

**Table 1. Access links and DOIs for data products used in this paper. N/A is short for not applicable.**

| No. | Name | Data access link | DOI | Reference |
|---|---|---|---|---|
| 1 | Seismicity catalogue | https://zenodo.org/records/15284376 | 10.5281/zenodo.15284376 | Riedel-Hornig et al., 2025 |
| 2 | 1P | https://service.iris.edu/fdsnws/dataselect/1/ | 10.7914/SN/1P_2018 | Wiens & Magnani, 2018 |
| 3 | C | https://service.iris.edu/fdsnws/dataselect/1/ | No DOI | Universidad de Chile, 1991 |

| 4 | C1 | https://service.iris.edu/fdsnws/dataselect/1/ | 10.7914/SN/C1 | Universidad de Chile, 2012 |
|---|---|---|---|---|
| 5 | CX | https://service.iris.edu/fdsnws/dataselect/1/ | 10.14470/PK615318 | GFZ, 2006 |
| 6 | G | https://service.iris.edu/fdsnws/dataselect/1/ | 10.18715/GEOSCOPE.G | IPGP y EOST, 1982 |
| 7 | GE | https://service.iris.edu/fdsnws/dataselect/1/ | 10.14470/TR560404 | GEOFON, 1993 |
| 8 | GT | https://service.iris.edu/fdsnws/dataselect/1/ | 10.7914/SN/GT | USGS, 1993 |
| 9 | IQ | https://service.iris.edu/fdsnws/dataselect/1/ | 10.14470/VD070092 | Cesca et al., 2009 |
| 10 | IU | https://service.iris.edu/fdsnws/dataselect/1/ | 10.7914/SN/IU | USGS, 1988 |
| 11 | VC | http://www.sernageomin.cl/ | No DOI | SERNAGEOMIN, 2015 |
| 12 | WA | https://service.iris.edu/fdsnws/dataselect/1/ | No DOI | UNSJ, 1958 |

**Author contribution**

M.RH., C.S. and A.T. gathered the necessary data from repositories and/or agencies. M.RH. with contributions by C.S. and J.P. designed the workflow and processed the data. M.RH. prepared the manuscript with contributions from all co-authors.

**Competing interests**

The authors declare that they have no conflict of interest

**Acknowledgments**

M. Riedel-Hornig has received funding from the Agencia Nacional de Investigacion y Desarrollo (ANID) grant 21220812. This project is also part of an international collaboration that has been cofounded by ANID project FOVI 240179 and FOVI

220075. A. Tassara acknowledges support from ANID FONDECYT project 1240862. C. Sippl received funding from the European Research Council (ERC) through the Horizon 2020 program (ERC Starting Grant MILESTONE; StG2020-947856):

C. Sippl and J. Puente were supported by the Czech Academy of Sciences through a Mobility Plus (MPP) Grant, grant number ANID-23-08. We would also like to thank OVDAS for kindly providing their data for us.

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
