# Peer review of "Seismicity catalogue of the entire Chilean margin (18° to 56°S) from an automated approach"

_Earth System Science Data, 2025_

## Author Response (AR1)

Reviewer 1:

This is an excellent manuscript and earthquake catalog for the years 2017 to 2021 spanning the Chilean subduction zone. This is a huge tectonic province with seismicity in the crust, on the plate boundary, and in the slab, and generation of a low completeness level, largely relocated in 3D structure, high quality catalog not plagued by network changes is a major contribution (of course, one hopes that the time span increases and there can be, say, five year updates going forward, as temporal patterns may be missed in the current catalog. The authors make many reasonable decisions of how to deal with the diverse sources of seismic signals and the vast along-strike length of the subduction zone, and these are clearly laid out. It seemed quite reasonable to include or exclude various data sources, and it is helpful to have it so clearly documented. The procedures applied to develop the catalog are widely utilized and appropriate for the catalog construction, with good description of the implementation of Earthquake Transformer, PyOcto, NonLinLoc and HypoDD, so the results are about the best that can be expected. The basic methods are not really unique or original, but the applications to the specific situation of Chile is. Some further lowering of completeness level might be possible for some subregions using template matching, but pushing the overall completeness down to magnitude 2.5 is already a major contribution and the catalog will be useful for many basic science applications. The catalog and phase picks are downloadable (already 106 downloads), and the map and cross-section figures show many intriguing features much more clearly than apparent in other catalogs.

The manuscript is very well written and the figures are excellent. I only caught one possible mistatement on line "135", where I believe "furthest" should be "closest", as duplicates are reduced by discarding the events in overlapping segments nearer to an edge. If that one statement is checked, I can recommend publication pretty much as is. This is a valuable contribution.

Author response:

Thank you for your kind words and positive comments about our work. We do hope to eventually extend the catalogue at least up to 2025, however, this will likely not be done right away, as: 1) The time needed to (re)process such a large data set does not fit into the time constrains of the PhD Thesis that the current catalogue and paper are involved in. Note that at least the relative relocation step needs to be re-done for all events, not only those newly added, whenever the catalogue is extended. 2) we currently do not have access to OVDAS waveform data between 2022 and 2025, which we hope to obtain in the future.

We checked the error you point to on line 135 and corrected it, along with a related mistake further down the text on line 137.

Reviewer 2:

In their manuscript "Seismicity catalogue of the entire Chilean margin (18 to 56 S) from an automated approach", Riedel-Hornig et al. present a new earthquake catalog covering the 5 year period from 2017 to 2021. With ~600,000 event, this catalog is an order of magnitude large than the routine CSN catalog, providing a detailed view into the seismicity of the Chilean margin. Overall, this is an excellent manuscript. The work is technically sound and the manuscript is well-written and contains all necessary detail. Below I provide a list of comments that should be addressed, however, all of them concern minor points. I recommend publication once these comments have been addressed.

We appreciate the positive comments and general opinion of the reviewer.

In addition, I wanted to bring up two suggestions. However, I believe both of them are out-of-scope for the manuscript at hand and might rather be of interest for future work. First, I think it would be of great value to extend the catalog to a longer duration. The years since 2021 have good coverage and even analyzing the more sparse data before 2017 with the same workflow would substantially improve upon existing catalogs. The longer duration would allow to systematically study long-term trends. Second, while the location accuracy in the presented catalog is good, it could be improved substantially by incorporating correlation-based differential times, providing detailed insights into fine-scale structures.

Like we stated above, in the reply to reviewer 1's comment, we do hope to eventually extend the duration of the catalogue, however, that is not yet possible if we want to keep the same station set. Upon creating an updated and extended version, we will also consider your suggestion of incorporating correlation-based differential times. Regarding your other, specific comments, please find replies to them below. We made the suggested modifications, some of which include changes to the catalogue. Therefore, we also updated the dataset in the repository.

Detailed comments:

- Line 81: I'd recommend using "consistent in time" instead of "constant in time" for the catalog. In contrast, the use of "constant" to refer to magnitude of completeness later in the same line seems appropriate.

- We made this suggested change.

- Line 84: While I agree on the aspect of temporal changes, being able to identify spatial changes would require uniform station coverage and noise conditions, which are not present here.

- You are correct. We removed the section referring to spatial changes in the text.

- Figure 1: To make part b more informative, I'd suggest sorting the stations by their latitude and scale it to the map on the left. This way, readers can see if certain regions have varying coverage over time.

- We changed the figure, now stations are sorted by latitude. We added a gray background to the graph to make colors stand out more.

- For the processing with HypoDD, I'd recommend to explicitly state that no cross-correlation times were used to avoid potential questions.

- We added this to the text on line 145.

- For the magnitude calculation, which attenuation function did you use? Is this the same attenuation function used by the CSN?

- We used the IASPEI recommended function. We added a mention of this in the text on line 153. The attenuation function used by the CSN is not reported.

- For the classification of events, I'm slightly concerned that the tolerance margins might in places be too small to capture the systematic deviation between your catalog locations and slab2. For example, in Figure 3 panel 9, there is a large swath of seismicity between 200 and 300 km that seems mislabeled. Similarly, other research found notable differences in depth between local catalogs and slab2, for example, Sippl et al. (2018), González-Vidal et al. (2023), and Münchmeyer et al. (2025).

- This is a difficult matter to solve. Like you mention, previous studies as well as our catalogue show a considerable deviation of observed hypocentral locations from Slab 2. However, slab2 is the most accepted global slab model as far as we are aware, this being the reason we chose to use it. Even so, some misclassifications deriving from its usage are likely inevitable. In hopes of reducing them, we slightly changed our classification procedure. Outerrise, crustal and intraslab events remain the same. For interplate events, rather than using a fixed distance to the slab we implement a margin based on the average depth errors of events within a certain area (cells of 0.25° by 0.25°) between a minimum margin above and below

the slab of 2.5 km and maximum of 10 km. This should increase the width of the area where events are classified as interface towards the outside of the network, but should not significantly alter it below the network, where there is little discrepancy between slab 2 and our events. However, the end result does not differ significantly from our original classification when regarded at larger scale. For intraslab events, most of the discrepancies arise at deeper depths, where events are often located far away from the slab2 slab surface. We unfortunately do not have a better classification method for these events. Of course we could attempt to fit a slab model with our hypocenters, but at least for these deepest events, our locations may be far off due to the network-station geometry (see our test in Figure 10). We do not believe that the offset from slab2 generally produces a large problem for intraslab events, as they tend to be deeper than Slab 2 (with some exceptions) and thus should not be mislabeled with our method. On Figure 3, panel 9 in particular, some intraslab events plotted above the slab are due to the swath width of events plotted in the profile, while the slab surface is calculated from a line at the center of the profile. Nonetheless, we added a sentence clarifying that the usage of Slab 2 might produce errors (lines 174-177 and 187).

- Line 174: Is this the number of associated picks or the total picks? Could you provide the total number of picks before association as well? The same applies to line 230.

- We added this information on lines 182-185 and 247.

- Line 178-180: The observation that the segments 20S-24S and 30S-34S are seismically more active than the region in between should be discussed in a more nuanced way. These segments have substantially denser station coverage than the 24S-30S segment, so the completeness will be lower. I'd suggest estimating completeness individually for each of the segments, truncating at the highest of the three completeness values and see if the change in seismic activity still is significant. As far as I know, the effect should remain intact, but it would be good to make a more solid argument for it.

- We added sentences discussing seismic productivity more in depth (lines 190-193. The matter of Mc is addressed further down the text (lines 219-229).

- For the same segments, I find the argument of incoming topography not particularly convincing. First, the less active segment in between hosts the subducting Taltal and Copiapó ridges, so it raises the question why these don't correspond to enhanced activity. Second, it seems unlikely that the incoming ridges change seismicity patterns over more than 400 km long segments.

- We removed the mention of incoming topography.

- Figure 2a: Could you add whiskers for the width of the profiles?

- We added the requested whiskers.

- Figure 2: There is a curious cluster of seismicity at depth north of 20S and around 68W. I checked the catalogs from Sippl et al (2018, 2023) but neither of the two report this cluster. It think it would be worthwhile commenting on this and validating whether this is a real feature or a processing artifact.

- We believe the cluster you refer to is a deep cluster of intraslab seismicity, with relatively low event density. Its location is in line with the slab geometry, which gives us confidence that it is not an artifact. Similar clusters show up further south in our catalogue (see figure bellow) as well as those of Sippl et al. (2018, 2023). Thus, we think that it is simply part of the same process that produces deep intraslab seismicity in other areas. It is true that at this particular latitude, the same cluster does not show up in previous work. However, at least for the Sippl et al., (2018, 2023) catalogues, there are some events in that area that get discarded during the processing with HypoDD, as they are too sparse to be considered clustered by the algorithm and thus do not get relocated. We think that they do show up in our result simply because we have a higher event density and therefore, these events do get relocated by HypoDD.

[Figure]

Figure: Latitudinal profiles at 19-22° S. Circle 1 highlights the mentioned cluster, while circles 2 and 3 highlight clusters at the same depth but further south, which do appear in previous catalogues.

- Figure 4, panel 3: There is a substantial swath of seismicity well within the oceanic mantle shown. This seems like a surprising feature to me, which is again absent from the Sippl et al catalogs. Again, I'd recommend to comment on this feature and verify whether it is an artifact and what might be causing it.

- This feature in particular seems to be a network artifact. Further north, at 20°S (see figure 1) there is a large concentration of stations. Should events close to the eastern border of the network get picks from a lot of these stations but not from any close to the networks edge, they get located as seen with the scattered group of events you mention. A similar location artifact occurs in panel 7, as demonstrated by our tests with Argentinian picks. We added mention of this on lines 192-195.

- In your assessment of absolute uncertainties you report higher errors in latitude than longitude. This is genuinely surprising to me, especially for out-of-network clusters towards the east. Generally, I'd expect uncertainty ellipses here to be elongated in EW directions (perpendicular to the network) and narrow in NS direction (parallel to the network). Could you explain this behaviour?

- We re-checked our data and found a formatting error at the end of the catalogue processing, where labels for latitude and longitude errors were inverted. Thus, this was simply a mixup, and in the corrected version, the reported uncertainty ellipses are indeed elongated E-W, as it should be expected. We fixed this mistake in the dataset as well as in the figures and text. Thanks for pointing this out!

- Figure 7: Please specify the time frame, i.e., "Number of events per ?".

- We added the requested information in the description of figure 7.

- For the segment 24S-30S, you compare to the González-Vidal et al. (2023) catalog. I've recently published a newer catalog for this region in Münchmeyer et al. (2025), using a superset of the data used by González-Vidal et al. (2023) and covering a longer duration. Given this catalog has higher resolution and substantially more events, I'd suggest updating the comparison. However, I admit that my catalog was only available as a preprint at the time of submission of this manuscript and I also don't want to push my own work too much, so I fully appreciate if you decide not to change this comparison.

- We were not aware of your paper at the time of writing of the manuscript. While your catalogue certainly has more events and covers a longer time interval than Gonzalez-Vidal et al. (2023), the large-scale geometry and appearance of the seismicity, which we compare in this figure, is not hugely different between these

two catalogs. We thus decided to keep the comparison in the figure as is, but added references to Münchmeyer et al. (2025) in the text.

- Line 322: The statement that the DSZ is continuous throughout the Chilean margin is an oversimplification. There are substantial along-strike variations in the DSZ, including regions with only a single plane, merging planes, or cross-cutting features, as shown, for example, in Figure 7 of Münchmeyer et al. (2025). I'd assume you could find similar features in fine cross-sections of your catalog. Therefore, I'd suggest to use a more careful statement here.

- We changed "appears to be continuous" to "be present along most of the margin between latitudes 18°S to 35°S" on lines 342-343